# Comparison of Imaging Radar Configurations for Roadway Inspection and Characterization

**DOI:** 10.3390/s23208522

**Published:** 2023-10-17

**Authors:** Mengda Wu, Laurent Ferro-Famil, Frederic Boutet, Yide Wang

**Affiliations:** 1IETR (Institut d’Électronique et des Technologies du numéRique), UMR CNRS 6164, University of Rennes, 35000 Rennes, France; frederic.boutet@univ-rennes.fr; 2ISAE-SUPAERO (Institut Supérieur de l’Aéronautique et de l’Espace), University of Toulouse, 31400 Toulouse, France; laurent.ferro-famil@isae-supaero.fr; 3CESBIO (Centre d’Etudes Spatiales de la Biosphère), University of Toulouse, 31400 Toulouse, France; 4IETR (Institut d’Électronique et des Technologies du numéRique), UMR CNRS 6164, Nantes Université, F-44000 Nantes, France; yide.wang@univ-nantes.fr

**Keywords:** synthetic aperture radar (SAR), ground-penetrating radar (GPR), ground-based SAR (GB-SAR), tomographic SAR (TomoSAR), roadways, artificial defects, bistatic radar, forward scattering, back scattering

## Abstract

This paper investigates the performance of a wide variety of radar imaging modes, such as nadir-looking B-scan, or side-looking synthetic aperture radar tomographic acquisitions, performed in both back- and forward-scattering geometries, for the inspection and characterization of roadways. Nadir-looking B-scan corresponds to a low-complexity mode exploiting the direct return from the response, whereas side-looking configurations allow the utilization of angular and polarimetric diversity in order to analyze advanced features. The main objective of this paper is to evaluate the ability of each configuration, independently of aspects related to operational implementation, to discriminate and localize shallow underground defects in the wearing course of roadways, and to estimate key geophysical parameters, such as roughness and dielectric permittivity. Campaign measurements are conducted using short-range radar stepped-frequency continuous-waveform (SFCW) devices operated in the C and X bands, at the pavement fatigue carousel of Université Gustave Eiffel, over debonded areas with artificial defects. The results indicate the great potential of the newly proposed forward-scattering tomographic configuration for detecting slight defects and characterizing roadways. Case studies, performed in the presence of narrow horizontal heterogeneities which cannot be detected using classical B-scan, show that both the coherent integration along an aperture using the back-projection algorithm, and the exploitation of scattering mechanisms specific to the forward-looking bistatic geometry, allows anomalous echoes to be detected and further characterized, confirming the efficacy of radar imaging techniques in such applications.

## 1. Introduction

Roadways represent an essential kind of infrastructure for transportation systems, and their safe and efficient operation is critical for society. However, over time, they can experience degradation due to a variety of factors, including climate, traffic loads, and material quality [1]. Invisible defects, caused by embedded and structural cracks, are serious issues that can lead to costly repairs or even accidents. Therefore, there is a need for effective techniques to evaluate the structural integrity and maintain the durability of roadways [2].

Microwave radar devices have proven to be valuable sensors for detecting and characterizing subsurface features. In particular, ground-penetrating radar (GPR) B-scan measurements have been extensively used for roadway inspection [3,4,5], using incoherent series of high-resolution vertical radar profiles sampled along scanning paths. Efficient techniques, based on such acquisitions, have been developed for characterizing the structure of complex underground media [6,7,8,9]. Advanced signal processing methods may be used to retrieve high-precision vertical information and estimate structural properties of the observed roadways [10,11].

Synthetic aperture radar (SAR) imaging is a coherent signal processing technique which uses spatial and spectral diversities in order to synthesize a 2-D map of the reflectivity of a scene [12]. SAR imaging is generally operated in a side-looking configuration, which allows important geophysical parameters such as roughness or humidity to be estimated using different modes of diversity [13]. Classical 2-D SAR mapping is affected by a cylindrical ambiguity and cannot discriminate scatterers located at different elevation positions and whose responses fall into the same azimuth-range resolution cell. Tomographic SAR (TomoSAR) imaging represents a solution to this limitation, and may be employed to estimate the 3-D reflectivity of environments [14,15]. This technique relies on the acquisition of multi-baseline interferometric SAR data from slightly shifted trajectories [14], which are coherently combined in order to generate 3-D images. TomoSAR has been used to investigate the characteristics and scattering mechanisms of complex 3-D media such as snow [16], ice [17,18], and forests [19]. Practical studies used near-range TomoSAR imaging to explore detailed structural properties by employing ground-based SAR (GB-SAR) systems [16,17,20,21].

In a typical back-scattering (BSC) configuration, TomoSAR imaging is usually implemented using several acquisition passes involving co-located transmitters and receivers. The length of the additional aperture determines the vertical resolution of the resulting 3-D image. This study also considers SAR tomography performed in a forward-scattering (FSC) configuration, i.e., with transmitting and receiving antennas located on opposite sides of the scene. In such a mode, the vertical resolution is fixed by the signal bandwidth, whereas the aperture length sets the resolution in the horizontal direction. FSC mechanisms are known to have a much higher energy than their BSC counterparts, due to the fact that waves travel along nearly specular paths. These facts make FSC SAR tomography a very interesting technique for the refined characterization of roadways.

This paper proposes a new FSC tomographic imaging mode and makes a comparative analysis with the traditional imaging configurations. This study shows that:FSC TomoSAR is well adapted for characterizing all roadway defects, and outperforms nadir-looking B-scan mode, particularly with improved resolution in the X band.The utilization of polarimetric diversity enables the estimation of dielectric permittivity and the compensation of refraction.Roughness is estimated using a scattering model, which benefits from the angular diversity of the side-looking configuration.

Three radar modes, nadir-looking B-scan, BSC TomoSAR, and FSC TomoSAR are described and their advantages and drawbacks are presented in Section 2. Measurements conducted with a GB-SAR operated in all three configurations over a pavement fatigue test circuit are described in Section 3. Section 4 discusses and compares the performances of the radar modes over pavement defects corresponding with three artificially debonded areas. Further analysis and the estimation of advanced roadway features, such as dielectric permittivity and roughness, are investigated in Section 5.

## 2. Investigated Imaging Modes

### 2.1. Nadir-Looking Ground-Penetrating Radar

A classical nadir-looking GPR system uses a pair of transmitting and receiving antennas to measure waves in the direction perpendicular to the air-ground interface, as illustrated in Figure 1. The vertical resolution δz of such an acquisition depends on the frequency bandwidth $B_{f}$, and is given by
(1)$$\delta z=\delta d=\frac{v}{2B_{f}}$$
where *v* is the propagation velocity of EM waves in the considered medium, and δd represents the range resolution. The azimuth resolution over distributed environments is given by the width of the Fresnel region of the antenna system, as explained in [4]. A nadir-looking GPR radar can be operated to acquire a B-scan which contains a series of echos at different positions along the scanning path. The ability of GPR devices to localize heterogeneities and discontinuities makes such systems well-adapted to applications related to road characterization and diagnosis.

Nevertheless, such a system has a very limited resolution in the azimuth direction, as B-scan is affected by migration effects, which reduces the possibilities of analysis. GPR systems being operated in the nadir direction generally do not exploit angular and polarization diversity.

### 2.2. Off-Nadir Synthetic Aperture Radar

Compared with a nadir-looking GPR achieving 1-D imaging along a track, an off-nadir SAR system utilizes a side-looking configuration to perform 2-D and 3-D focusing using horizontal and vertical spatial diversity, as shown in Figure 2. The back-projection algorithm is used to perform 3-D tomography in order to discriminate embedded cracks and to characterize dispersive media [15,22].

#### 2.2.1. Single-Channel Monostatic SAR

BSC is the natural observation mode of monostatic SAR, and provides measurements that are sensitive to roughness and polarization diversity. In the case of horizontal layers, the resolutions of a single-channel BSC image are given by
(2)$$\delta z_{B}{|}_{y=y_{0}}=\frac{\delta d}{\cos\theta_{1}},\;\delta y_{B}{|}_{z=z_{0}}=\frac{\delta d}{\sin\theta_{1}}$$
where $\left(y_{0},z_{0}\right)$ represent the ground-range and elevation coordinates corresponding to $\theta =\theta_{1}$.

BSC imaging suffers from a deterioration in horizontal resolution at the near range, i.e., when $\theta_{1}$ reaches values close to 0 in (Equation 2), and is subject to a cylindrical ambiguity. Cylindrical ambiguity is illustrated in Figure 3, with scatterers located at Cartesian coordinates $y_{i}$ and $z_{i}$ which cannot be separated in a BSC configuration, whatever the signal bandwidth, as they are located on a cylinder, centered at the radar position whose radius verifies $d_{1}=\sqrt{{\left(H-z_{i}\right)}^{2}+y_{i}^{2}},\;\;\forall i$. This ambiguity is a strongly limiting factor for the characterization of volumetric media.

#### 2.2.2. Single-Channel Bistatic SAR

In order to meet the requirements of road inspection, the FSC mode of bistatic SAR is proposed to enhance the longitudinal detection capability [23]. Forward-scattered signals traveling along a quasi-optical specular path are much more energetic than back-scattered ones, significantly improving the signal-to-noise ratio (SNR) and detection capability. The range resolution δd is converted into quasi-constant vertical resolution $\delta z_{F}$, whereas the ground-range resolution $\delta y_{F}$ is very coarse
(3)$$\delta y_{F}{|}_{z=z_{0}}=\frac{2\delta d}{\left|\sin\theta_{1}-\sin\theta_{2}\right|},\;\delta z_{F}{|}_{y=y_{0}}=\frac{2\delta d}{\left|\cos\theta_{1}+\cos\theta_{2}\right|}\approx \sqrt{2}\delta d\;\forall y$$
where the expression is given at a certain case with incident and reflecting angles of $45^{\circ }$.

Considerable information in the vertical direction can be obtained from a single channel, i.e., with only one pair of antennas. One may note that in such a configuration, the estimated ground-range location is ambiguous, as illustrated in Figure 4. Indeed, considering a scatterer lying on a horizontal layer, there are always two ground-range locations, $y_{1}$ and $y_{2}$, leading to equally traveled FSC distances, $d\left(y_{1}\right)=d\left(y_{2}\right)$, with
(4)$$d\left(y\right)=d_{1}\left(y\right)+d_{2}\left(y\right)$$

#### 2.2.3. Tomographic Monostatic SAR

A synthetic aperture in the vertical direction can be introduced to implement BSC tomographic imaging, which allows the cylindrical ambiguity to be solved. The tomographic vertical resolution component related to angular diversity only is called $\delta z_{BT\infty }$ as it corresponds to the value obtained under the assumption of an infinite signal bandwidth, or δd≪δ⊥ in Figure 5a. The expression of the vertical resolution is given by
(5)$$\delta z_{BT\infty }=\frac{\lambda_{c}\sin\theta_{1}}{4\sin(\Delta \theta /2)}$$
where $\lambda_{c}$ is the carrier signal wavelength.

Computing effective resolution values may be required to account for the signal bandwidth as δd and δ⊥ reach comparable levels.

#### 2.2.4. Tomographic Bistatic SAR

Tomography is introduced in FSC mode to resolve the ambiguity problem in the range direction of a single-channel acquisition. The displacement of the transmitting antennas in the vertical direction synthesizes an antenna aperture, which focuses an equivalent beam, as indicated by the transparent red zone in Figure 2. The corresponding ground-range resolution, $\delta y_{FT}$, is given by
(6)$$\delta y_{FT}=\frac{\lambda_{c}}{2\cos\theta_{1}\sin\left(\Delta \theta /2\right)}$$

The features of the discussed imaging modes are summarized and compared in Table 1. GPR operated in the nadir-looking configuration achieves great resolution in the vertical direction but has limited discrimination in the azimuth direction. Tomography brings improved resolution and solves the ambiguity in both the BSC and FSC modes, at the cost of increased complexity. By implementing a single channel, FSC can obtain comparable vertical discrimination while keeping a simple structure.

## 3. Experiment Configuration for Roadway Characterization

### 3.1. Ground-Based SAR System Operated in BSC and FSC Configurations

A GB-SAR system operated in the BSC and FSC configurations, and depicted in Figure 6b, has been developed at the IETR, University of Rennes [24,25], in order to acquire data in both BSC and FSC modes at different frequency bands.

The SAR device is based on a vector network analyzer (VNA) having four bidirectional ports connected to horn antennas. The VNA generates stepped-frequency continuous waveforms (SFCWs) over a wide spectral domain and measures the response of the scene with a high accuracy. Three antennas are fixed on the platform facing the scene to be illuminated, whereas the other one is placed on the other side of the scene in order to acquire signals in FSC mode. The set of three antennas can be moved along a 2-D aperture in azimuth and elevation with a high-accuracy servo system and a high-precision lifting platform. In the present study, the system is operated in the C and X bands in both the BSC and FSC modes, and at HH and VV polarizations. The experiment parameters are summarized in Table 2, where $N_{f}$ is the number of frequency steps, and $L_{x}$ and $L_{z}$ are the aperture lengths in the azimuth and vertical directions, respectively.

As explained in [21,26], by combining the transmitting and receiving channels together with an adequate physical antenna spacing and platform motion in the vertical direction, it is possible to generate a quasi-uniform equivalent array of antennas in elevation, with an effective inter-element spacing in elevation denoted by dz in Table 2. The values of the equivalent spacings $dz_{B}$ and $dz_{F}$ differ in the BSC mode, as all three antennas on the same side are used as transmitters and receivers.

### 3.2. Experiment Setup and Scene Description

Measurement campaigns were carried out at the roadway test center of IFSTTAR, Nantes, France. The fatigue-test facility consists of a ring of full-scale asphalt paved road constructed with four load-bearing robotic arms rolling in a circle in order to create naturally deteriorated road conditions. One section of a 25 m loop road is built to simulate debonded areas by inserting artificial defects between two asphalt layers. Three types of materials, sand, geotextile, and air gap, are used to construct thirteen test zones, as shown in Figure 7; zones I11, I12, and I13 are associated with this experiment.

As shown in Figure 8a,d a layer of sand is inserted to simulate a light defect at zone I11 [28]. A serious defect is created at I12 using a rectangular patch of geotextile in Figure 8b,e. Figure 8c,f represent a defect having the shape of a horizontal crack, corresponding here to an air gap. It is important to know that the defects represent very shallow layers with a thickness of a few millimeters buried at a depth of approximately 8 cm below the upper ground surface, as indicated in Figure 8g.

The experiment is implemented with two groups of the nadir-looking GPR and the GB-SAR operated in BIZONA mode, as shown in Figure 6. The nadir-looking configuration uses a pair of antennas moved in the *X* direction to obtain an incoherent B-scan with vertical resolution δz. Off-nadir GB-SAR configurations synthesize a vertical aperture $L_{z}$ by lifting the antenna array from 0.56 m and 0.64 m to 1.36 m, for the C and X bands, respectively. The spatial resolutions obtained for different configurations are calculated using the experimental setup illustrated in Table 2 at the scene center, i.e., with $\theta_{1}=45^{\circ }$. These calculations consider angular apertures of $\Delta \theta =10^{\circ }$ and $\Delta \theta =20^{\circ }$, in both the C and X bands. The detailed results provided in Table 3 confirm that FSC mode maintains a constant vertical resolution $\delta z_{FT}$, independent of the angular aperture, which surpasses the resolution $\delta z_{BT}$ in BSC mode with a full angular aperture. On the other hand, the BSC mode reaches more refined ground-range resolution values than the FSC mode. Increasing the angular aperture allows the vertical resolution to be improved in BSC mode and the horizontal one in FSC mode.

## 4. Measurement Campaign Results

### 4.1. Comparison of Configurations

The imaging results are given for all the investigated modes are given in Figure 9 for nadir-looking profiling, in Figure 10 for BSC 2-D, in Figure 11 for BSC tomo, in Figure 12 for FSC single-channel, and in Figure 13 for FS tomo, over the three defect zones of geotextile, sand, and air gap.

Over the geotextile zone, the nadir-looking incoherent profile measured in the C band and depicted in Figure 9a, shows a strong echo for the air–asphalt interface, and a lower intensity one corresponding to the geotextile layer. The peak of the geotextile response is not located at the actual interface depth of 8 cm due to the slower propagation of electromagnetic waves in the host medium. By taking the dielectric permittivity of asphalt, $\varepsilon_{r}\approx 5$, as reported in [27], into account, the depth axis can be scaled by $\sqrt{\varepsilon_{r}}$, shifting back the response of the layer to approximately 20 cm. The profiles measured in the C and X bands exhibit a comparable ratio between the reflectivities of the ground surface and of the defect layer. With the dielectric properties being relatively stable over the microwave region in dry conditions [29], the linear increase in wave attenuation with the signal frequency is somehow compensated for by the larger reflectivity of the measured interfaces, which appear rougher at shorter wavelengths. One may note that due to the larger bandwidth of the acquired signals, the vertical resolution in the X band is much more refined than the one observed in the C band. Over sand and air-gap areas, in Figure 9b,c, the echo from the defect has a very weak amplitude in the C band due to a lower reflectivity with respect to the geotextile layer. In the X band, the echo is almost invisible, and cannot be localized.

The BSC 2-D results, i.e., classical SAR imaging in BSC mode, illustrated in Figure 10a–c, show that the expected structure of the underlying ground medium cannot be appreciated using this kind of measurement due to the aforementioned cylindrical ambiguity. One may notice the strong decrease in reflectivity in the ground-range direction, due to both the change in incident angle and the antenna pattern of the system. The strong reflectivity area located at the ground-range position of 2.5 m corresponds to the response of the BIZONA acquisition mast depicted in Figure 6b. No reliable information can be extracted from this kind of intensity image for defect discrimination in either the C or X bands.

On the other hand, the results obtained from BSC tomography, i.e., Figure 11a–c, possess interesting features in the X band for all the measured media. One can neatly discriminate an upper interface with a strong reflectivity and a lower one corresponding to the defect layer. The response of the underlying defect is focused along a bent shape. This phenomenon is due to the refraction effect, as explained in [17,20,21]. Interestingly, the application of the same processing steps to the C-band data leads to a lack of response from volume. This mis-focusing is due to the decorrelation between the echos. Indeed, in the C band, the roadway appears as a very smooth surface; most of the emitted energy is reflected away in the forward direction, and the back-scattered signals have a small amplitude, and hence a non-adapted SNR. In such conditions, SAR tomography gives very poor results as the signal cannot be correctly focused.

The single-channel FSC results shown for the geotextile layer in Figure 12a lead to conclusions that are similar to those raised for the nadir-looking experiments. A strong reflection is observed from the underlying defect in the C band, whereas a weaker response is measured in the X band. Concerning sand and air-gap areas, the detection of the defect layer is extremely difficult in Figure 12b,c. This result is interesting as it shows that, at lower frequency, it is possible to detect underlying defects using a single pair of Tx and Rx antennas with 1-D focusing in distance only.

As shown in Figure 13a, tomographic FSC is well adapted to the detection of underlying structures, as it measures strong scattering contributions from both the air–asphalt interface and the subsurface defect at both frequencies in the case of the geotextile zone. The responses of the defects remain significant in the C band for the sand and the air gap, and are still perceivable in the X band, as depicted in Figure 13b,c. One may note the reduced resolution in the C band compared with in the X band, and also the absence of speckle patterns in FSC mode. This is due to the nature of scattering mechanisms in the forward direction, which are composed of coherent terms related to specular reflections.

### 4.2. Tomography Correction with Refraction Compensation

Electromagnetic waves exhibit different propagation characteristics in free space and soil. At the interface of different layers, a refraction phenomenon occurs and the propagation velocities vary according to the permittivities of the media [30,31]. The positions and shapes of underground targets may be distorted if calculating the distance directly. To focus images accurately, the propagation trace and distance must be corrected and compensated for.

In a 3-D Cartesian coordinate system where the antenna’s phase center is located at (0,0,H), the received signals $S_{i,f}$ of the *i*th point-like scatterer in the frequency domain can be expressed as
(7)$$S_{i,f}=s_{i}e^{-j4\pi fd_{i}/c}$$
where $s_{i}$ represents the complex echo amplitude and *f* is the working frequency. The equivalent distance from the antenna to the *i*th point scatter $\left(x_{i},y_{i},z_{i}\right)$ is denoted by $d_{i}$.

Under common free-space conditions, this distance can be directly calculated using the Euclidean distance. However, for underground detection, the propagation trace is not straightforward due to the refraction phenomenon. To simplify the model, the geometry configuration is represented using 2-D (ρ,z) coordinates, where $\rho =\sqrt{x^{2}+y^{2}}$ indicates the horizontal distance. The antenna is set at (0,H) and the interface is at z=0, as shown in Figure 14. The equivalent distance $d_{i}$ is given by
(8)$$d_{i}=d_{air}+d_{soil}=\sqrt{\rho_{i0}^{2}+H^{2}}+n_{r}\sqrt{{\left(\rho_{i}-\rho_{i0}\right)}^{2}+z_{i}^{2}}$$
where the refraction coefficient $n_{r}=\sqrt{\varepsilon_{2}/\varepsilon_{1}}$, and the horizontal distance of the refraction point is $\rho_{i0}$, which is estimated by a binary searching algorithm in the range of $\left(\rho_{il},\rho_{ih}\right)$.

During the focusing, the distance between the antenna and each pixel in the focusing zone needs to be recalculated as the antenna moves in both the azimuth and elevation directions. The amount of computation increases dramatically due to the refraction compensation incorporated in each calculation. In order to optimize the computational flow, an array of equivalent distance d is calculated as a function of (ρ,z)

for a given antenna height *H*
(9)d=f(ρ,z|H)
where (d,ρ,z) is the set of $\left(d_{i},\rho_{i},z_{i}\right)$. The equivalent distance is calculated only once for each antenna pass. For an imaging grid of 13 × 800 × 10,000 pixels, the improved compensated approach takes 62% less time, as compared in Table 4.

The influence of refraction on tomography and its compensation are illustrated using signals simulated for a specific configuration shown in Figure 15a and the system configuration described in Section 3.2. The ground layer is located at elevation 0 and the defect layer is buried in the asphalt at a depth of 8 cm. The asphalt is considered as a homogeneous medium whose dielectric permittivity is equal to 5, which is equivalent to a refractive index, $n_{r}$, of 2.2.

The uncompensated focusing of the simulated signals in the BSC configuration is shown in Figure 15b. The upper interface is correctly represented as a straight stripe at z=0 over the ground range specified in the simulation configuration. Whereas the depth of the defect layer is miscalculated, being larger than the actual depth and varying according to the ground-range value. This non-linear delocalization effect has been illustrated and explained in previous works [20,21]. In the FSC configuration, the delocalization has a larger magnitude due to the specific geometry configuration of bistatic acquisition, depicted in Figure 15d. And the originally straight buried layer appears as bent on the resulting tomogram. One can observe on the compensated tomograms of Figure 15c,e, that the proposed compensation techniques successfully relocate the buried layer at its original location.

The effectiveness of this reconstruction technique is verified over the geotextile area in both BSC and FSC modes in the X band, as shown in Figure 16. The conclusion raised in using simulated signals also applies in this case, that from the corrected tomogram, we can estimate the actual depth of the defect layer, and again in the focusing accuracy. In FSC mode, the response of the buried layer appears right below the upper layer after compensation in Figure 16d. Whereas in an uncompensated tomogram, Figure 16c, it appears slightly shifted to larger ground values. The application of refraction compensation corrects the position and geometrical features of underground layers.

## 5. Result Analysis and Advanced Feature Extraction of Roadways

### 5.1. Analysis of Imaging Configurations

Over geotextile zones, corresponding to severely debonded roadways, nadir-looking GPR and FSC allow the defect layer to be discriminated in both the C and X bands. In BSC tomographic mode, the use of X band signals reaches a satisfying performance too. As illustrated in Figure 9a and Figure 12a, nadir- and forward-looking configurations can both lead to high-level results, even when using a single channel only.

In more complex situations involving sand defects, the nadir-looking mode requires the use of lower-frequency signals in order to obtain a measurable response from the sand layer, whose response is not observable in the X band. In the C band, echos are not well-localized and may be confused with the sidelobe of the ground interface, as shown in Figure 17a. In FSC mode, the tomographic scattering intensity is high in the C band and is still detectable in the X band, as depicted in Figure 17b. Interestingly, Figure 17c indicates that the sand layer behaves as a rough surface in the X band, and leads to significant scattering in BSC mode, although it is affected by speckle.

Over the air-gap zone, a sidelobe and discontinuities in the host medium bring confusion in the analysis of the nadir-looking results, even in the C band, as indicated in Figure 18a. In comparison, the air gap can be accurately localized in FSC mode in both the C and X bands, as displayed in Figure 18b. Figure 18c also shows a back-scattering source of reflectivity at the location of the air gap, with a lower amplitude compared with the sand defect due to a reduced apparent roughness.

In conclusion, FSC provides satisfactory features over all three debonded areas in the C and X bands. Nadir-looking mode operated with a moderate bandwidth works well for geotextiles, but in the case of light defects, such as sand and air gaps, the results indicate a reduced detection potential. Interestingly, roughness-induced features can be measured in BSC mode over all defects in the X band.

### 5.2. Permittivity Estimation with Polarization Response

Unlike nadir-looking mode, side-looking configurations such as FSC and BSC may be used to naturally estimate the roughness and moisture of rough surfaces or slightly rough surfaces. Indeed, the radar response of rough surfaces or interfaces observed at the incidence angle is sensitive to the polarization of the incoming wave. The sensitivity, which is modulated by the dielectric permittivity of the observed medium, is enhanced by moisture content. On the other hand, the analysis of the response of surfaces observed at different incidence angles may be used to estimate their roughness property. In the considered cases, the small-perturbation model (SPM), whose application is valid when observing roadway surfaces in both the C and X bands, may be used to efficiently simulate the reflectivity of such environments [13,32]. This model has the ability to characterize the roughness-dependent part of the response from the part related to the dielectric properties.

The expression of the scattering coefficient $\sigma_{pq}$ for the polarimetric channels *p* and *q*, which are equal to *v* or *h*, is given by
(10)$$\sigma_{pq}=8{\left|k^{2}\sigma_{h}\cos\theta_{1}\cos\theta_{2}\alpha_{pq}\right|}^{2}W\left(k_{x}\right)$$
where $W\left(k_{x}\right)=\frac{1}{2}l^{2}\exp\left(-k_{x}^{2}l^{2}/4\right)$ is the power spectrum function, *p* and *q* indicate the polarization directions *V* and *H*, *k* is the wave number, $\sigma_{h}$ denotes the surface height standard deviation, and *l* is the correlation length.

The polarization amplitude $\alpha_{pq}$ can be represented as a function of the relative dielectric constant $\varepsilon_{r}$, the incidence angle $\theta_{1}$ and the scattering angle $\theta_{2}$, and for the relative magnetic permeability $\mu_{r}=1$, as
(11)$$\begin{array}{cc} \alpha_{HH} & =\frac{-\cos\phi (\varepsilon_{r}-1)}{\left(\cos\theta_{2}+\sqrt{\varepsilon_{r}-\sin^{2}\theta_{2}}\right)\left(\cos\theta_{1}+\sqrt{\varepsilon_{r}-\sin^{2}\theta_{1}}\right)}\\ \alpha_{VV} & =\frac{\left(\varepsilon_{r}-1\right)\left(\sqrt{\left(\varepsilon_{r}-\sin^{2}\theta_{2}\right)\left(\varepsilon_{r}-\sin^{2}\theta_{1}\right)}\cos\phi -\varepsilon_{r}\sin\theta_{1}\sin\theta_{2}\right)}{\left(\varepsilon_{r}\cos\theta_{2}+\sqrt{\varepsilon_{r}-\sin^{2}\theta_{2}}\right)\left(\varepsilon_{r}\cos\theta_{1}+\sqrt{\varepsilon_{r}-\sin^{2}\theta_{1}}\right)} \end{array}$$
where ϕ=0 in FSC mode, whereas ϕ=π and $\theta_{1}=\theta_{2}$ in BSC mode. The cross-polarization coefficient $\alpha_{HV}=0$ at order 1.

From the expression of the reflectivity model by SPM given in (Equation 10) and (Equation 11), one realizes that the components related to the dielectric permittivity $\varepsilon_{r}$ of the scattering coefficient can be isolated by computing the ratio $\frac{\sigma_{HH}}{\sigma_{VV}}$, which is equal to the polarization ratio $|\frac{\alpha_{HH}}{\alpha_{VV}}{|}^{2}$. For known incidence and scattering angles, the estimated dielectric permittivity is the one that fits the ratio value. This approach is run on 3-D refraction-compensated tomograms, as shown in Figure 19a,c,e, where the black boxes indicate the limits of the areas used to estimate the reflectivity values, with $L_{1}$ and $L_{2}$ corresponding to the asphalt and defect layers, respectively. The ratio values are calculated based on the angles $\left(\theta_{1},\theta_{2}\right)$ corresponding to the maximum point inside the box.

The estimated permittivity values of asphalt over the three zones, as indicated in Table 5, fall within the commonly encountered range for road asphalt [33,34,35] and are consistent with results reported in [27] over the same test circuit. The similar $\varepsilon_{r}$’s of asphalt over the sand and air-gap zones are estimated from measurements acquired during the same sunny day, whereas over the geotextile zone, the measurement took place after a rain event, resulting in a high estimated permittivity value. It is known that the dielectric permittivity of buried defects can be modulated by humidity related to preceding rain events, with water infiltrating and remaining for a long time within the cavity of the defect. This is certainly the reason for the extremely high permittivity value of the geotextile, which was soaked with water, with $\varepsilon_{r}\gg 80$.

Interestingly, this approach could be implemented by a single channel without focusing, as shown in Figure 19b,d,f. The reflectivity is calculated with the summation along two curves, corresponding to the responses of the asphalt and defect layers. The dielectric permittivity is estimated under the certain condition, assuming that the targets are located in the middle of the transmitting and receiving antennas in the ground-range direction. Similar results are obtained and compared in Table 5, except the ratio value of $L_{2}$ in the sand zone, where the reflectivity is small in single-channel mode, and 3-D focusing allows the SNR value to be enhanced by combining channels. Hence, single-channel observations are not sufficient for defects with very small responses.

### 5.3. Roughness Estimation

The roughness features of both the ground surface and the embedded sand layer are estimated using the aforementioned SPM scattering model and imaging results obtained in the X band using the VV-polarization channel. The BSC mode is selected due to its well-known sensitivity to roughness properties when operated at different incidence angles. The expression for the scattering coefficient given in (Equation 10) can be rewritten in BSC mode, with $k_{x}=2k\sin\theta_{1}$ and $\theta_{1}=\theta_{2}$, as
(12)$$\sigma_{vv\_BSC}=4{\left(k\sigma_{h}\right)}^{2}{\left(kl\right)}^{2}\cos\theta_{1}^{4}{\left|\alpha_{VV}\right|}^{2}\exp\left[-{\left(kl\sin\theta_{1}\right)}^{2}\right]$$
where the roughness parameters are estimated using a fit between the expression of (Equation 12) and average profiles estimated from the imaging results for different incidence angles. As one may note, the surface height standard deviation $k\sigma_{h}$ contributes only to the amplitude of $\sigma_{vv\_BSC}$, and cannot be estimated with our system, as it is not absolutely calibrated. The correlation length kl is, hence, estimated by normalizing the simulated profiles to give the best fit to the measured ones. This procedure is applied to the 2-D and 3-D focusing results after compensating for the attenuation related to the distance between the radar and the considered 3-D location.

For the ground surface, one may observe in Figure 20c that, despite the fact that the 2-D focusing results are from the sum of both the ground surface response and that of the underlying sand layer, the scattering features are similar to those obtained in 3-D imaging when selecting only the response of the ground surface. This is due to the fact that the response of the sand layer is significantly smaller than that of the ground surface. The 3-D imaging can then be used to characterize the apparent roughness of the sand layer by applying the SPM model at the corresponding depth, with $\theta \in [30^{\circ },60^{\circ }]$, and the fitting results are shown in Figure 20d. The obtained correlation length values are 3.5 for the ground surface and 4 for the sand layer, indicating slightly rough surfaces.

## 6. Conclusions

This paper compares the classical nadir-looking B-scan GPR mode with the newly proposed BSC TomoSAR and FSC TomoSAR configurations for the characterization and diagnostic determination of roadways. Measurements conducted with a GB-SAR operated in both the C and X bands over a pavement fatigue carousel are used to evaluate the ability of the investigated configurations for detecting artificial defects corresponding to very thin layers, or cracks, filled with a geotextile, sand, or with air.

The analysis outcomes reveal distinct performance characteristics for each configuration:The nadir-looking configuration, without ultra-large bandwidth conditions, fails to detect defects having a weak response, like sand-filled cracks or air gaps, in both the C and X bands.The BSC TomoSAR mode accurately identifies geotextile and sand defects but cannot detect smooth air gaps due to insufficient response.The proposed FSC TomoSAR mode successfully isolates all defect types, benefiting from significant scattered signal amplitudes, even in the X band.

The results obtained in the X band show an improved resolution that helps to discriminate the defects. The gain in performance observed with the SAR-based configuration is due to the coherent integration of several range profiles to create an image, which maximizes the signal to noise ratio at the true location of the scattering source.

The use of polarimetric and angular diversity with side-looking configurations allowed us to:Estimate the dielectric permittivity of the layers;Correct geometric distortions induced by the refraction of the signal when propagating through the 3-D medium;Estimate some roughness indicators for the imaged interfaces using a scattering model.

The experimental results and theoretical analysis proposed in this paper demonstrate the advantages and inconveniences of the proposed configurations and methodologies, and their ability to detect defects hidden inside the roadway pavement.

One may note that a direct operational implementation of the FSC tomographic mode for roadway characterization is strongly limited by different factors. Compared with the nadir-looking B-scan, the FSC tomographic mode requires a more complex hardware configuration, with several antennas, receivers, transmitters, etc. The coherent processing, required to synthesize a 3-D image, is also more complex and needs to accurately calibrate the system. The acquisition time is also more important in the BSC and FSC configurations with respect to the classical B-scan.

Nevertheless, the authors are currently finalizing an adaptation of the FSC tomographic mode to operational constraints, which brings down the operational and computational complexities, as well as the acquisition duration, to the level of those experienced with B-scan, while maintaining the superior capabilities of this mode for detecting and characterizing roadway defects.

## Figures and Tables

**Figure 1 sensors-23-08522-f001:**
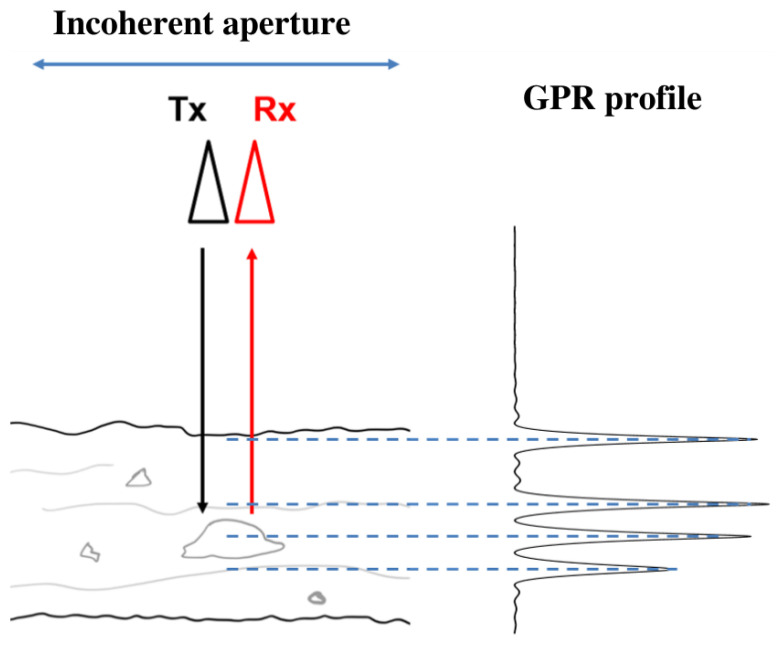
Geometry of a nadir-looking GPR acquisition.

**Figure 2 sensors-23-08522-f002:**
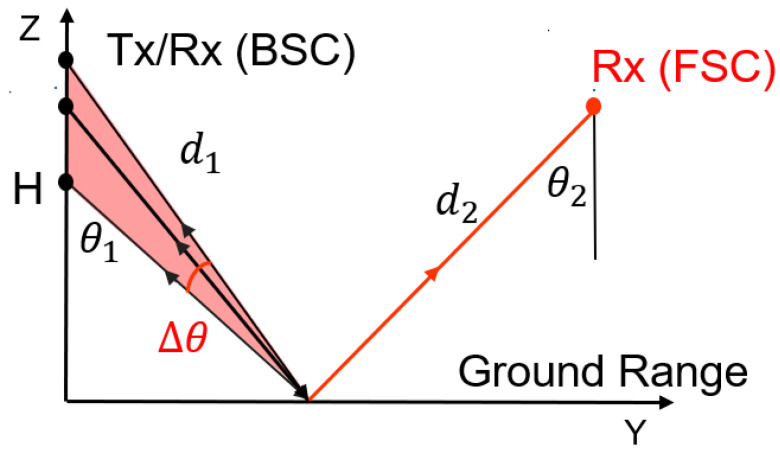
Geometry of off-nadir SAR imaging in BSC and FSC modes.

**Figure 3 sensors-23-08522-f003:**
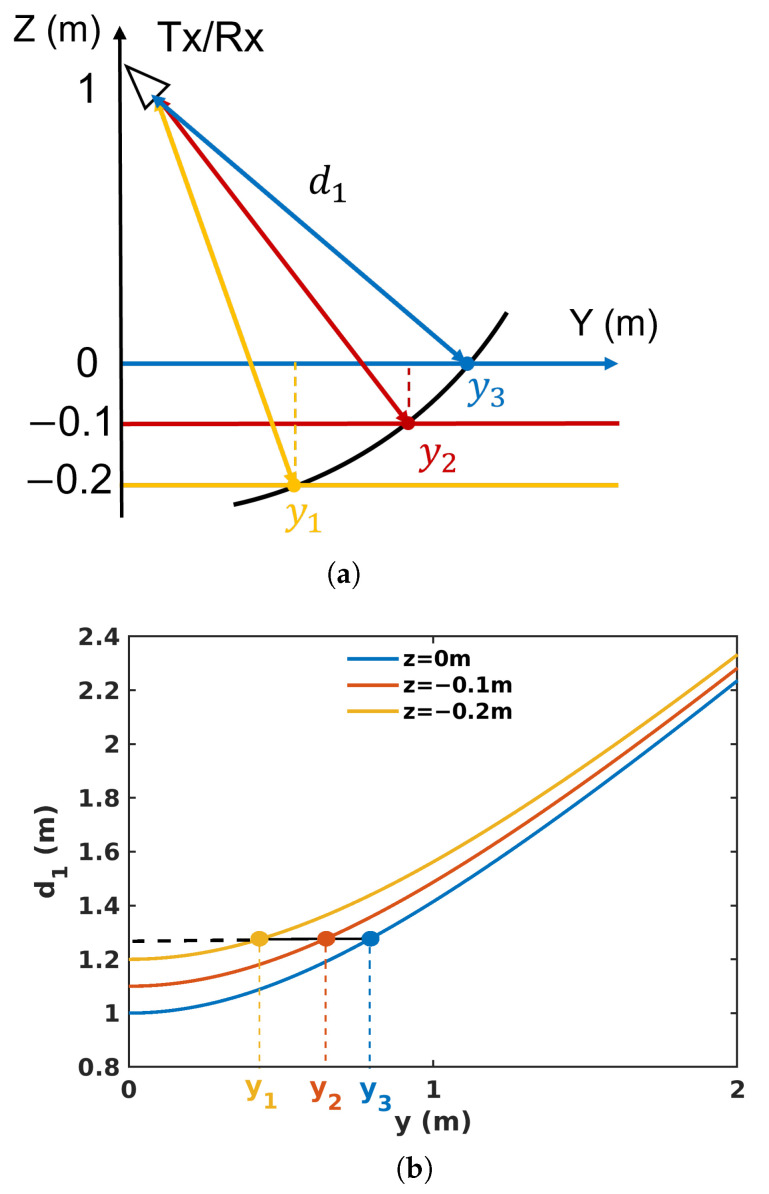
Illustration of cylindrical ambiguity when measuring 3 superimposed layers in BSC mode (**a**) in the ground-range elevation plane (**b**) in the ground-range distance plane.

**Figure 4 sensors-23-08522-f004:**
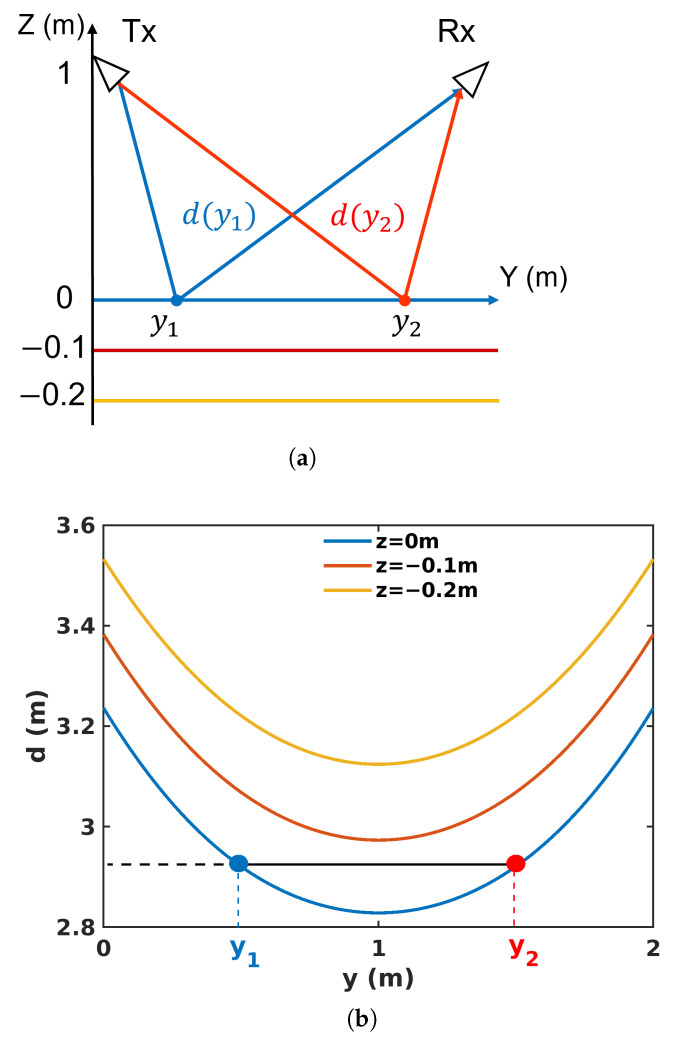
Illustration of ground−range ambiguity when measuring 3 superimposed layers in FSC mode (**a**) in the ground-range elevation plane, and (**b**) in the ground-range distance plane.

**Figure 5 sensors-23-08522-f005:**
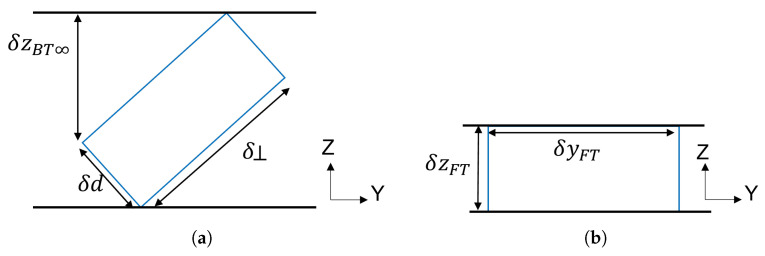
Graphical representation of tomographic resolution cells separating two horizontal layers in (**a**) BSC and (**b**) FSC modes with $\theta_{1}=\theta_{2}=45^{o}$.

**Figure 6 sensors-23-08522-f006:**
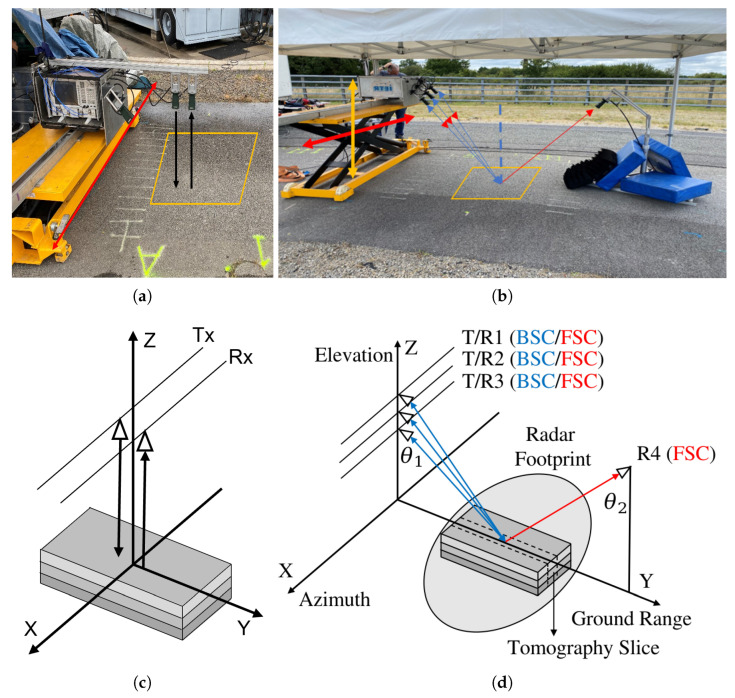
Configurations used during the experiment: (**a**) GPR system operated in nadir-looking configuration; (**b**) GB-SAR system operated in BSC and FSC configurations; (**c**) nadir-looking acquisition geometry; (**d**) BSC and FSC acquisition geometries.

**Figure 7 sensors-23-08522-f007:**
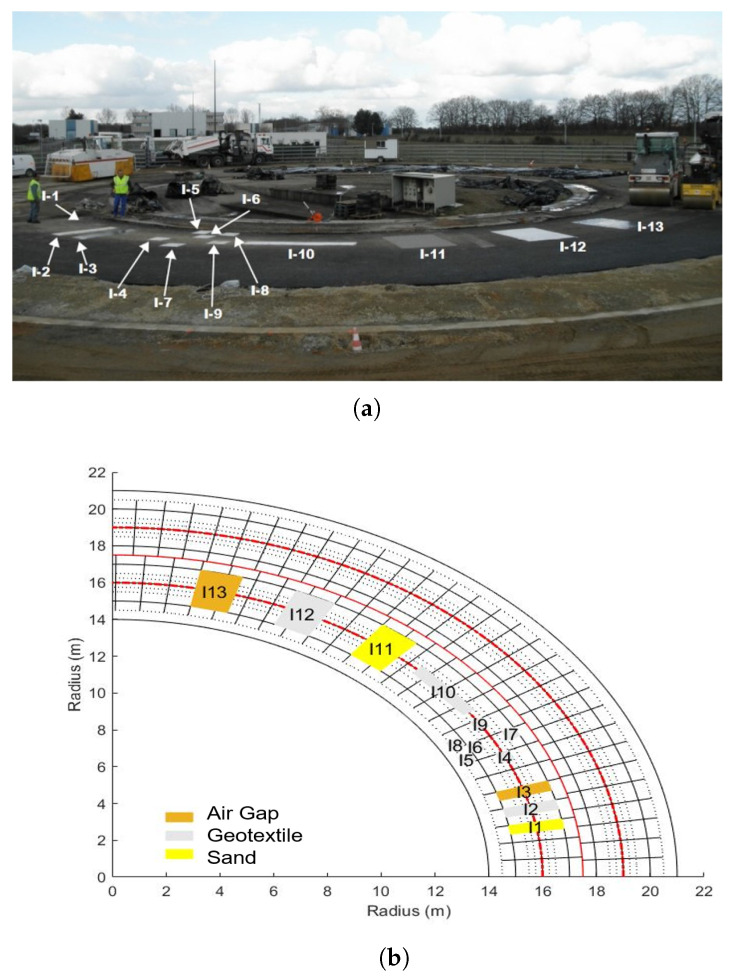
(**a**) Overview of the fatigue test circuit during the preparation of the defectuous roadway sections; (**b**) locations of defects on the circuit ring [27].

**Figure 8 sensors-23-08522-f008:**
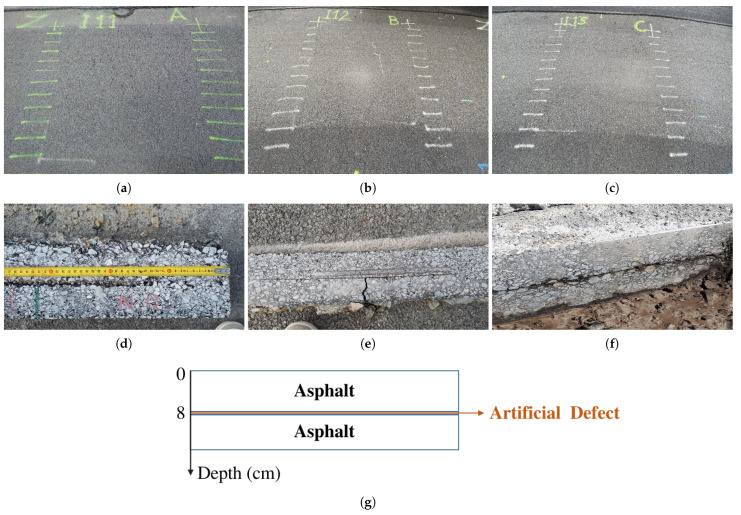
Description of the debonded roadway zones from (**a**–**c**) top view and (**d**–**f**) side view, over (**a**,**d**) sand-defect zone, (**b**,**e**) geotextile-defect zone, and (**c**,**f**) air-gap zone [27]. (**g**) Side-view structure of a debonded zone.

**Figure 9 sensors-23-08522-f009:**
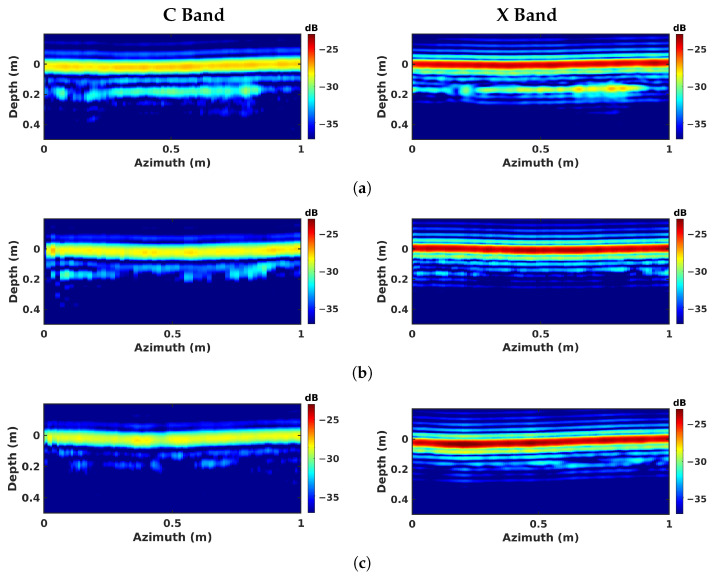
Nadir-looking focusing results obtained over different zones: (**a**) geotextile zone, (**b**) sand zone, and (**c**) air-gap zone.

**Figure 10 sensors-23-08522-f010:**
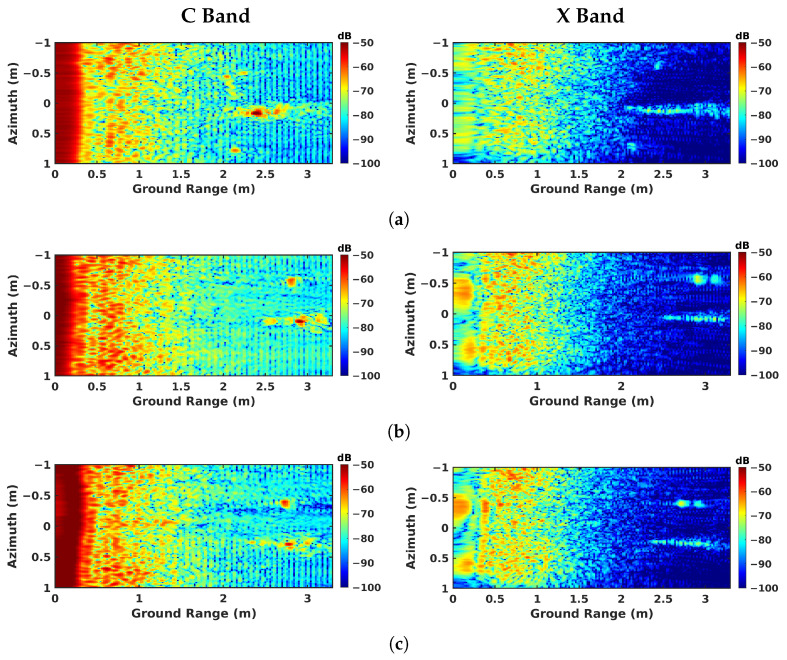
BSC 2-D focusing results obtained over different zones: (**a**) geotextile zone, (**b**) sand zone, and (**c**) air-gap zone.

**Figure 11 sensors-23-08522-f011:**
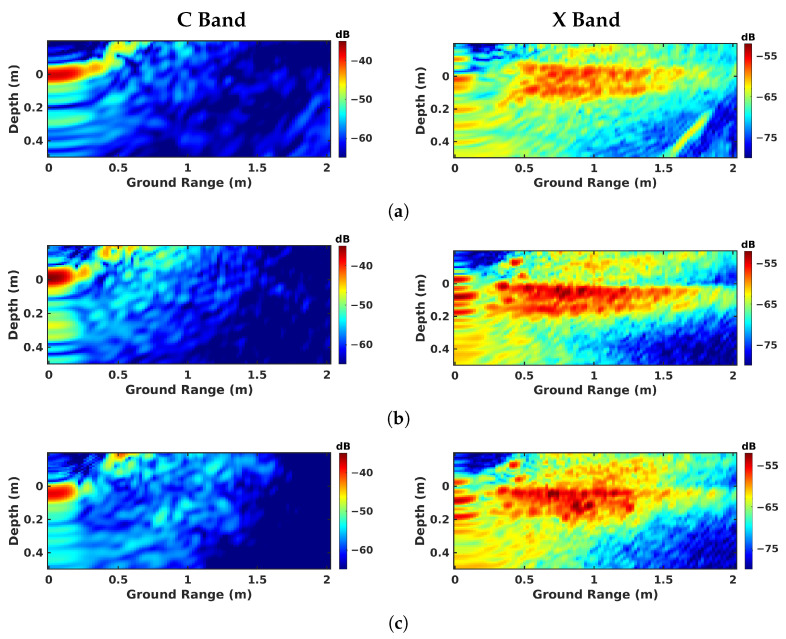
BSC tomographic imaging results over different zones: (**a**) geotextile zone, (**b**) sand zone, and (**c**) air-gap zone.

**Figure 12 sensors-23-08522-f012:**
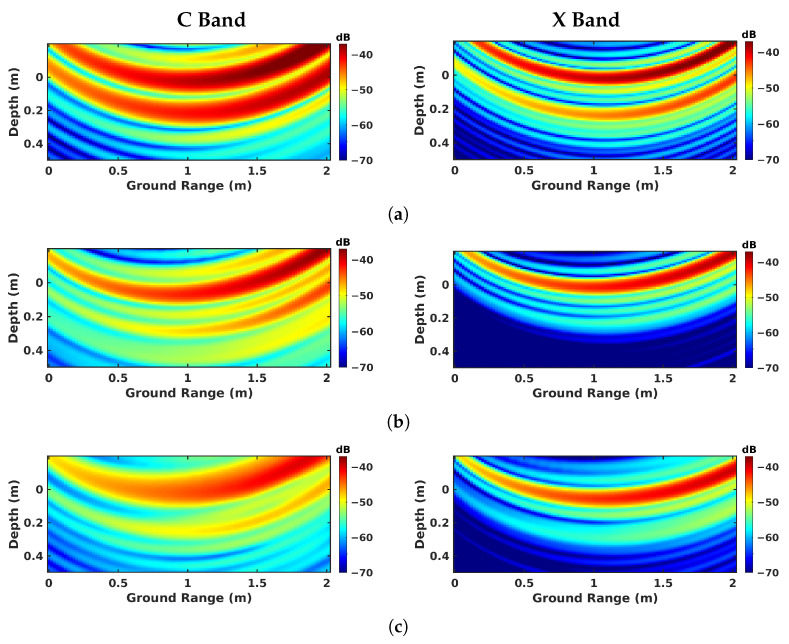
FSC single-channel focusing results over different zones: (**a**) geotextile zone, (**b**) sand zone, and (**c**) air-gap zone.

**Figure 13 sensors-23-08522-f013:**
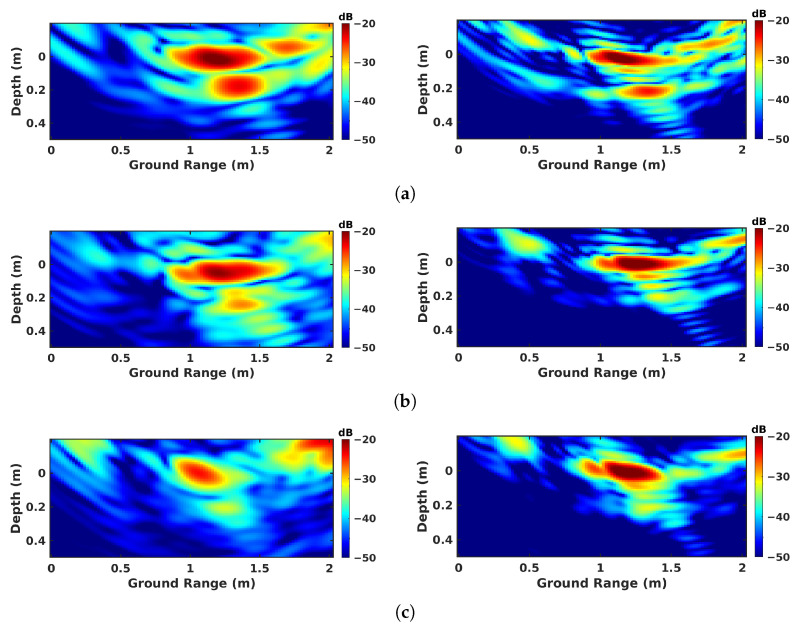
FSC tomographic imaging results over different zones: (**a**) geotextile zone, (**b**) sand zone, and (**c**) air-gap zone.

**Figure 14 sensors-23-08522-f014:**
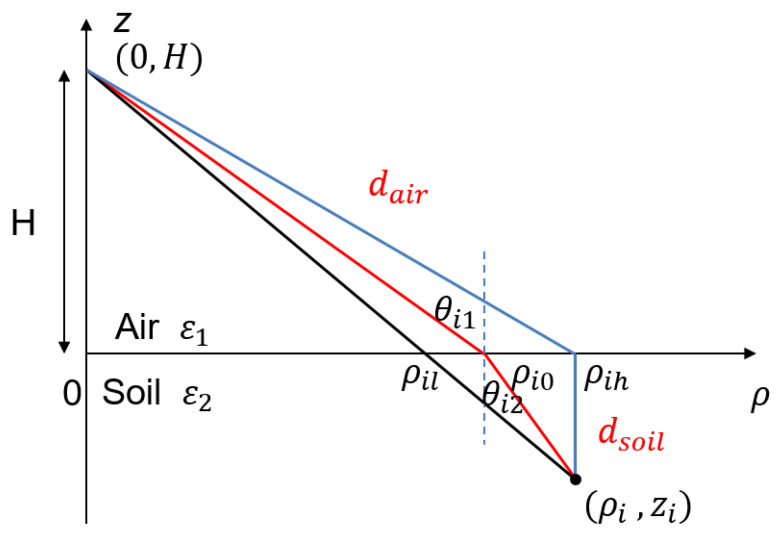
Schematic of refraction compensation.

**Figure 15 sensors-23-08522-f015:**
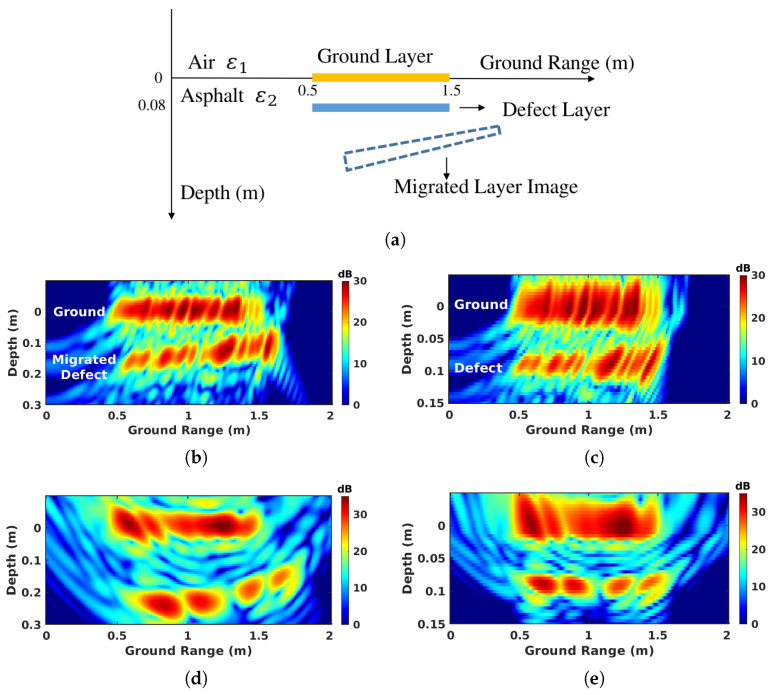
Illustration of refraction effect and its compensation with (**a**) schematic description of the simulated medium, using simulation results: (**b**) uncompensated BSC Tomo, (**c**) compensated BSC Tomo, (**d**) uncompensated FSC Tomo, and (**e**) compensated FSC Tomo.

**Figure 16 sensors-23-08522-f016:**
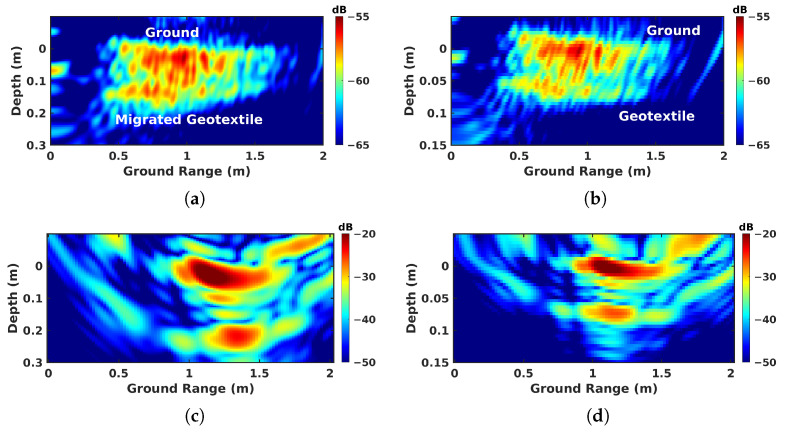
Results comparison before and after compensation over the geotextile zone in the X band: (**a**) uncompensated BSC Tomo, (**b**) compensated BSC Tomo, (**c**) uncompensated FSC Tomo, and (**d**) compensated FSC Tomo.

**Figure 17 sensors-23-08522-f017:**
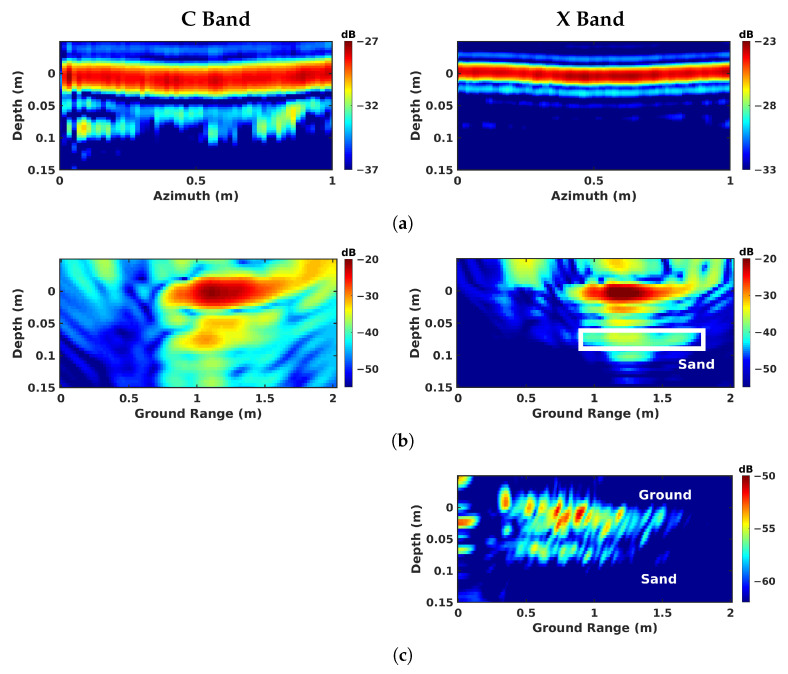
Compensated imaging results over the sand zone: (**a**) nadir-looking, (**b**) FSC Tomo, and (**c**) BSC Tomo.

**Figure 18 sensors-23-08522-f018:**
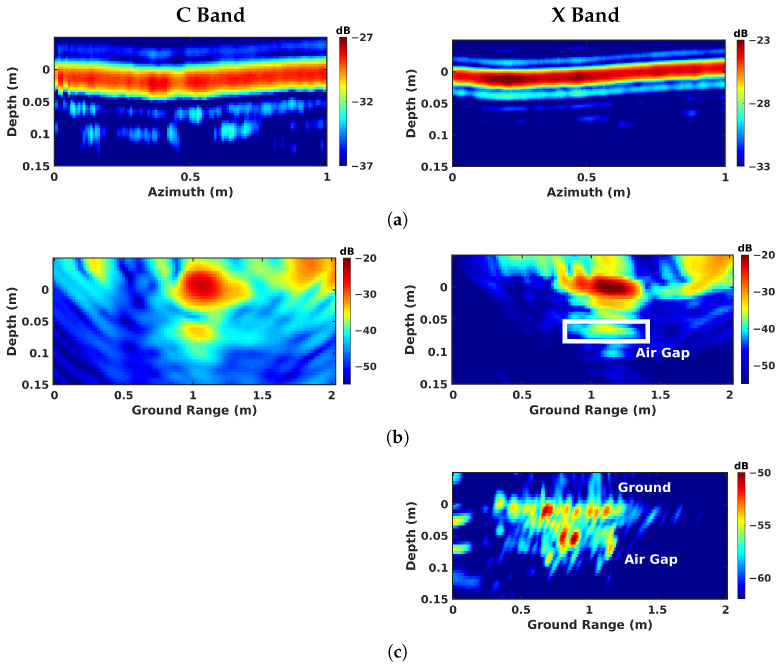
Compensated imaging results over the air-gap zone: (**a**) nadir-looking, (**b**) FSC Tomo, and (**c**) BSC Tomo.

**Figure 19 sensors-23-08522-f019:**
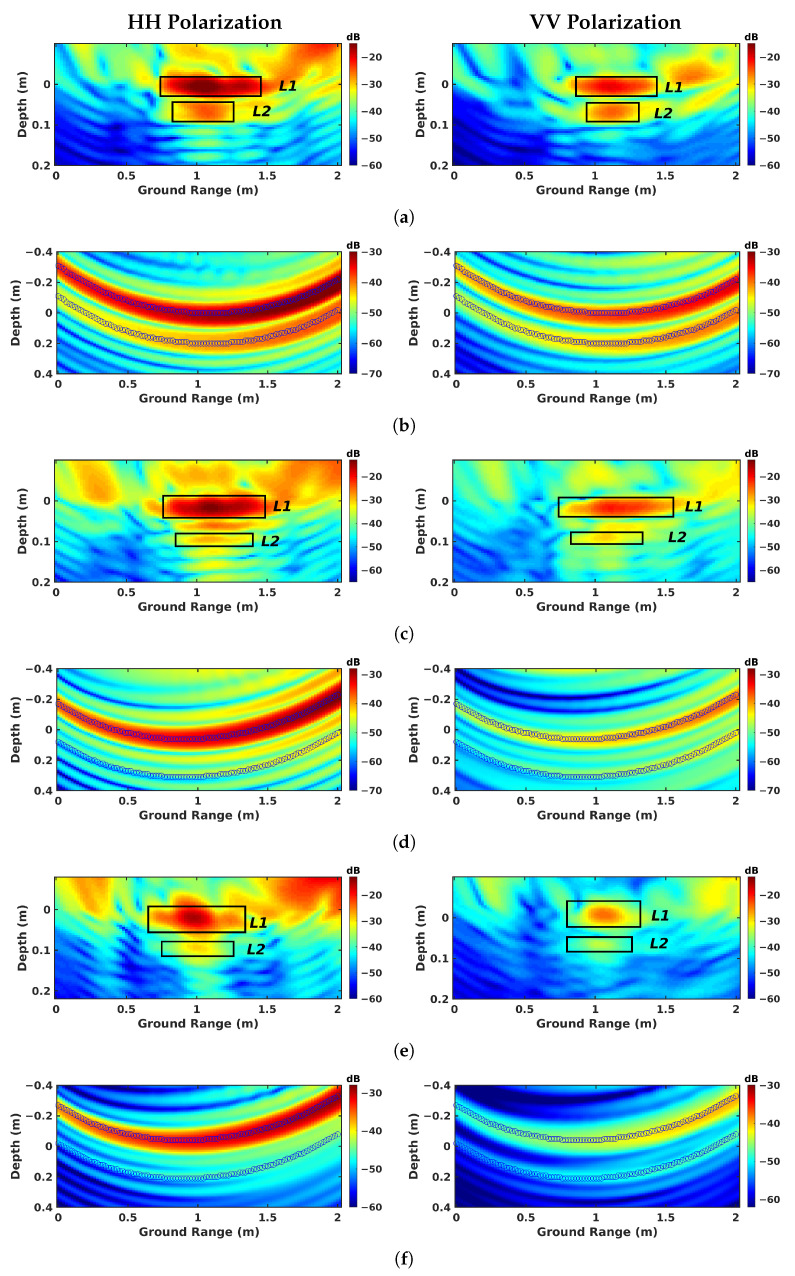
Parametric reflectivity of different zones: (**a**,**b**) geotextile zone, (**c**,**d**) sand zone, and (**e**,**f**) air-gap zone, in (**a**,**c**,**e**) FSC Tomo mode, and (**b**,**d**,**f**) FSC single-channel mode. $L_{1}$: ground asphalt; $L_{2}$: defect layer.

**Figure 20 sensors-23-08522-f020:**
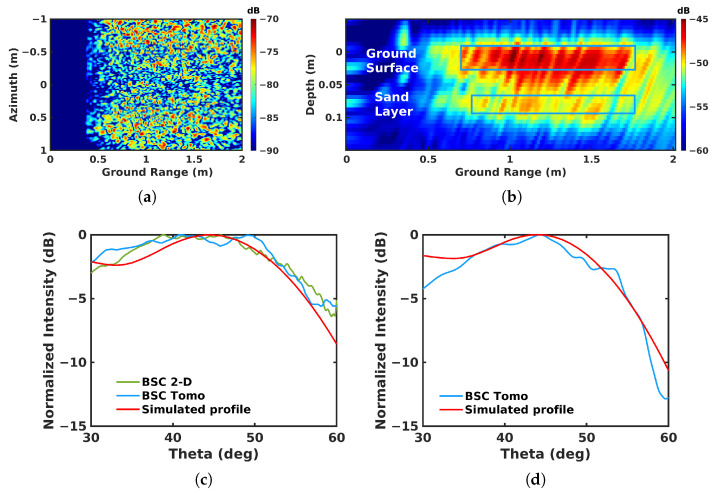
Roughness estimation: (**a**) distance-compensated BSC 2-D image; (**b**) distance-compensated BSC tomogram; (**c**) measured and simulated profiles over the ground surface; (**d**) measured and simulated profiles over the sand layer.

**Table 1 sensors-23-08522-t001:** Comparison of imaging modes.

Imaging Modes	Advantage	Drawback
Nadir-looking	Excellent vertical resolution	No focusing on AZ, low polarimetric sensitivity
BSC Tomo	Polarimetry, good resolution, sensitive to roughness	Complex, long time
FSC Tomo	Polarimetry, good resolution both H/V	Complex, long time
FSC single-channel	Polarimetry, good vertical resolution	Poor horizontal discrimination

**Table 2 sensors-23-08522-t002:** GB-SAR system parameters.

	C Band	X Band
$f_{c}$	5.85 GHz	10 GHz
$B_{f}$	2.3 GHz	4 GHz
$N_{f}$	1001	2001
$L_{x}$	2 m	2 m
$L_{z}$	0.8 m	0.72 m
$dz_{B}$	0.04 m	0.03 m
$dz_{F}$	0.08 m	0.06 m
Polarization	HH/VV	HH/VV

**Table 3 sensors-23-08522-t003:** Special resolutions for $\theta =45^{\circ }$ and considering propagation in free space (unit: cm).

		$\Delta \theta =10^{\circ }$				$\Delta \theta =20^{\circ }$			
	δd	$\delta z_{BT}$	$\delta y_{BT}$	$\delta z_{FT}$	$\delta y_{FT}$	$\delta z_{BT}$	$\delta y_{BT}$	$\delta z_{FT}$	$\delta y_{FT}$
C Band	6.52	15.01	9.22	9.22	41.57	9.82	9.22	9.22	20.85
X Band	3.75	8.73	5.3	5.3	24.32	5.71	5.3	5.3	12.21

**Table 4 sensors-23-08522-t004:** Comparison of computation time over 13 × 800 × 10,000 pixels.

Focusing Method	Time
Raw focusing without compensation	8 s
Focusing with compensation	82 s
Focusing with optimized compensation	36 s

**Table 5 sensors-23-08522-t005:** Comparison of polarization ratio and estimated permittivity of experimental scenes.

Zone	Geotextile Zone		Sand Zone		Air-Gap Zone	
Marker	$L_{1}$	$L_{2}$	$L_{1}$	$L_{2}$	$L_{1}$	$L_{2}$
FSC Tomo $\frac{\sigma_{HH}}{\sigma_{VV}}$	6.26	0.29	7.55	2.9	8.15	4.63
$\varepsilon_{r\_tomo}$	5.8	≫80	4	9.5	4.1	6.3
FSC Single-Channel $\frac{\sigma_{HH}}{\sigma_{VV}}$	6.16	0.26	8.85	0.07	9.72	3.17
$\varepsilon_{r\_single}$	6.1	≫80	3.7	≫80	3.7	7.8

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
