# Peer review of "Comparison of Imaging Radar Configurations for Roadway Inspection and Characterization"

_sensors, 2023, doi:10.3390/s23208522_

Round 1

Reviewer 1 Report

The article shows the improvement of GPR data for shallow detection of thin defects in pavements with the use of different techniques and survey techniques.

The language used is good and the order and organization is fine.

Overall, the article demonstrates well the advantage of the TomoSAR technique for defects detection. Despite this, the reviewer does not see, now, how this can be used alone for road survey.

The reviewer suggests the article to be published after addressing the following suggestions/comments.

Comments:

The description of the methods could be improved, especially in terms of explaining the use of certain terms not usual in the PGR field such as “nadir-looking”, as well as a better and simpler description of the cylindrical and range ambiguities in terms of SAR.

The reviewer understands the results from the “nadir-looking” but what is the advantage/difference relatively to normal GPR normal operating procedures?

In terms of the use of SAR, it seems that the use of such technique is to further investigate defects previously detected by GPR, or how the procedure can be used to detect defects?

No references to Figures 7a,b,c were given in the text.

Nothing special.

Author Response

We would like to express our sincere gratitude to the reviewer for the valuable feedback and insightful comments. All the formulated comments have been thoroughly addressed in this response document, and the paper has been revised accordingly. Our responses are outlined in blue as in the attachment, and the modifications in the paper are highlighted in red.

Reviewer 2 Report

This paper explores the application of 3 radar imaging modes for inspecting and characterizing roadways, including nadir-looking B scans, back-scattering tomographic SAR and front-scattering tomographic SAR. Comparison experiments were conducted with a C-band radar and an X-band radar on 3 types of defect situations, showing the advancement of the proposed method. Polarimetric and angular diversities are also used to estimate the dielectric permittivity and the roughness indicators of the road layers. The content of the article is substantial, but there are still some issues to fix before the possible publication.

1.Abstract: I suggest that, in the abstract section, the authors should emphasize the focused problem of the article and the advantages of the proposed method compared with the existing ones.

2.Figure 3: Figure 3 shows the cylindrical ambiguity geometry. In the above figure, I suppose the 3 lines in different colors should be of the same length d1. Please modify the figure. In the below figure, according to the description in lines 101 to 103, I suppose cylindrical ambiguity means a fixed d1 corresponding to different values of y. Please clarify this.

3.Figure 5: The authors used some words to describe the directions in the geometry, such as azimuth, vertical, elevation and so on. It would be clearer for the readers if these directions are marked in Figure 5.

4.Figure 8, 9, 10: The captions of the 3 figures are the same, please distinguish them. From the color bars, I noticed the color limits(ranges) are different in every sub-figures. I suggest the authors should control the color limits(ranges) of sub-figures the same.

5.Section 4.1: When discussing the results, different sub-figures need to be compared in Figure 8, 9, 10. I suggest adjusting the figures' order and put 8(a), 9(a), 10(a) in a single figure, and 8(b), 9(b), 10(b) as well.

6.Table 3: Please add some discussions about the resolution results in Table 3.

7.Conclusion: The conclusion should discuss the limitations of the study, such as experimental conditions, data size etc. The conclusion should explain how these factors affect the results. Besides, the conclusion should offer some suggestions for future research, such as how to extend or deepen the study of this paper.

Author Response

We would like to express our sincere gratitude to the reviewer for the constructive remarks and perceptive comments. Every comment has been carefully addressed in this response document, leading to comprehensive revisions in the paper. In the subsequent sections, our responses are presented in blue in the attachment, and the modifications in the paper are highlighted in red.

Round 2

Reviewer 2 Report

The authors have addressed all my concerns. I suggest accepting this paper after fixing some typos as below.

1. Line 10: Please give the complete form of the abbreviation ‘SFCW’ since it appears in the paper for the first time.

2. Formula (12): Is the symbol αVV the same as the formula (11)? If so, please keep the case of subscripts consistent, uppercase or lowercase.

3. Please maintain consistency of the use of short horizontal lines in abbreviations in the paper. For instance, ‘3D’, ‘3-D’, ‘B scans’, ‘B-scans’, ‘TomoSAR’, ‘Tomo-SAR’……

Author Response

Dear Reviewer,

We greatly appreciate your careful review and the positive recognition of our manuscript.

We have corrected typographical errors and ensured consistent use of horizontal lines in abbreviations, such as: TomoSAR, 2-D, 3-D, B-scan.

Thank you for your time and consideration.